# Adoption and validation of the European Association for the Study of the Liver algorithm for the noninvasive diagnosis of advanced fibrosis in metabolic dysfunction-associated steatotic liver disease in low-resource South Asian settings

Chamila Mettananda[1,2]*, Chamila Ranawaka[3], Thimira Egodage[1], Channaka Dantanarayana[1], Rumal Fernando[1], Lakmali Ranaweera[3], Dulani Kottahachchi[4], Shirom Siriwardhana[5], Arunasalam Pathmeswaran[6], Anuradha Dassanayake[1,2], Janaka de Silva[2,7]

1 Department of Pharmacology, University of Kelaniya, Ragama, Sri Lanka, 2 Professorial Medical Unit, North Colombo Teaching Hospital, Ragama, Sri Lanka, 3 Department of Gastroenterology, North Colombo Teaching Hospital, Ragama, Sri Lanka, 4 Department of Physiology, University of Kelaniya, Ragama, Sri Lanka, 5 Department of Anatomy, University of Kelaniya, Ragama, Sri Lanka, 6 Department of Public Health, University of Kelaniya, Ragama, Sri Lanka, 7 Department of Medicine, University of Kelaniya, Ragama, Sri Lanka

* chamila@kln.ac.lk

## Abstract

### Introduction

Patients with significant liver fibrosis (SF) are likely to progress to advanced chronic liver disease (ACLD). Therefore, liver-directed therapy is indicated. The European Association for the Study of the Liver-2024 (EASL) recommends annual screening of patients with diabetes for SF/ACLD using the FIB-4 score, followed by vibration-controlled transient elastography (VCTE) in patients with FIB-4 ≥ 1.3 in ≤65-year-olds and ≥2.0 in > 65-year-olds. Because VCTE is not freely available in resource-limited settings, we revised the EASL algorithm to prioritise referrals for VCTE in such settings and validated it in an external cohort.

### Methods

We conducted a cross-sectional study of adults with type 2 diabetes (T2DM) and ultrasonographic evidence of steatotic liver disease (SDL) attending three outpatient medical clinics in the Gampaha District, Sri Lanka. FIB-4 scores were calculated, and those with scores ≥1.3 underwent VCTE. SF was defined as liver stiffness measure (LSM) ≥ 8.0 kPa. Factors significantly associated with SF/ACLD were identified using multiple logistic regression (LR). We then developed a new criterion for VCTE referral and compared the number of referrals for VCTE when using the new criterion-based

**Data availability statement:** The Ethics Review Committee (ERC) of the Faculty of Medicine, University of Kelaniya, has not given the approval to share participants' data from the Ragama Health Study, as the initial study conducted in 2007 had not obtained Ethical approval for data sharing. Therefore, access to the" data could be granted for a valid request to the corresponding author or the ERC (ercmed@kln.ac.lk).

**Funding:** This study was supported by the Grant RC/2024/03 of the University of Kelaniya, Sri Lanka, awarded to Chamila Mettananda. The funders had no role in the study design, data collection and analysis, the decision to publish, or the preparation of the manuscript.

**Competing interests:** The authors have declared that no competing interests exist.

**Abbreviations:** SF, Significant Fibrosis; ACLD, Advanced Chronic Liver Disease; EASL, European Association for the Study of the Liver; VCTE, Vibration-Controlled Transient Elastography; T2DM, Type-2 Diabetes Mellitus; FIB-4 Score, Fibrosis-4 Index; LSM, Liver Stiffness Measure; LR, Logistic Regression; OR, Odds Ratio; BMI, Body Mass Index; MASLD, Metabolic Dysfunction-Associated Steatotic Liver Disease; AF, Advanced Fibrosis; SLD, Steatotic Liver Disease; USS, Ultrasonography; AST, Aspartate Aminotransferase; ALT, Alanine Aminotransferase; CAP, Controlled Attenuation Parameters; PPV, Positive Predictive Value; NPV, Negative Predictive Value; SD, Standard Deviation; IQR, Interquartile Range; CI, Confidence Interval; P, Probability; CV, Cardiovascular.

algorithm with the EASL algorithm. We validated the new criterion-based algorithm in an external cohort of 372 patients with MASLD.

## Results

We studied 363 patients, and 128 had an FIB-4 score of ≥ 1.3. Of them, 121 underwent VCTE, and 76 had an LSM ≥ 8.0 kPa. On multivariable LR, VCTE-diagnosed SF/ACLD was independently associated with diabetes of ≥ 5 years duration (OR 3.8, p = 0.035), micro/macrovascular complications (OR 19.4, p = 0.016), and BMI of ≥ 25 kg/m2(OR 6.2, p = 0.003). We revised the VCTE referral criterion as "patients having EASL FIB-4 criterion plus one or more of the three other factors: diabetes of ≥ 5 years duration, presence of micro/macrovascular complications or BMI≥25 kg/m2,". The number of VCTE referrals indicated using the EASL algorithm was 96, compared to 83 with the new criterion, resulting in a 13.5% reduction. In the external validation cohort, the new algorithm reduced the number requiring VCTE referral by 15.5%.

## Conclusions

Adopting the new criterion for VCTE referral in patients with MASLD appears more cost-effective for detecting SF/ACLD in low-resource settings in South Asia.

---

## Introduction

Metabolic Dysfunction-Associated Steatotic Liver Disease (MASLD) is the most common liver disease globally, with around 25–30% prevalence [1–4]. It is a spectrum of diseases ranging from simple steatosis through steatohepatitis to hepatic fibrosis, cirrhosis and hepatocellular carcinoma [5]. Liver-related morbidity and mortality of MASLD depend on the stage of liver fibrosis [6,7]. Only around 10% of people with MASLD progress through significant fibrosis (SF) to advanced chronic liver disease (ACLD) (encompassing advanced fibrosis (AF) and cirrhosis) or hepatocellular carcinoma [8,9]. The only curative treatment for end-stage liver disease is liver transplantation, which is not freely available [10]. Therefore, early detection of patients at the stage of SF and initiation of liver-directed therapies to prevent disease progression to AF and end-stage liver disease is important [11–16].

The gold standard for staging of liver fibrosis is liver biopsy. But, as it is an invasive procedure with a morbidity and mortality, the current recommendation is to measure the stage of liver fibrosis non-invasively using methods like vibration-controlled transient elastography (VCTE) using adapted thresholds; Liver stiffness measurements(LSM) of ≥8 kPa and ≥ 12 kPa using VCTE are suggestive of SF and ACLD, respectively [17–19]. However, MASLD is common, and screening with elastography in the whole population is not possible. Therefore, the European Association for the Study of the Liver (EASL) has developed an algorithm for screening patients with MASLD using a two-step method [20]. It recommends screening of patients with type-2 diabetes(T2DM), obesity plus one or more of cardiometabolic risk factors or persistently elevated liver enzymes

for steatotic liver disease (SLD) first with Fibrosis-4 (FIB-4) score and then performing VTCE in patients likely to have SF who are at risk of progressing to ACLD. The latest 2024 update of the EASL algorithm recommends VCTE in patients with a FIB-4 score ≥1.3 in patients < 65 years or a FIB-4 score ≥2.0 in patients older than 65 years [13]. Patients with FIB-4 < 1.3 are assumed to be at low risk of AF. Patients with a FIB-4 ≥ 2.67 are at high risk of advanced fibrosis or cirrhosis FIB-4 ≥ 1.3 (or ≥2.0 in individuals aged >65 years), have an intermediate risk for AF, and are recommended to have VCTE to detect the stage of fibrosis exactly [13]. However, VCTE availability is very limited in resource-limited settings like South Asia, especially where the prevalence of diabetes is reaching epidemic proportions [21]. Sri Lanka also has limited availability of VCTE and is a country with a high incidence and prevalence of MASLD, obesity and T2DM [22–25].

Therefore, we aimed to identify predictors of SF/ACLD in a cohort of South Asians to further prioritise VCTE referrals beyond EASL guidelines.

## Methods and analysis

We conducted a cross-sectional study at three medical/endocrine outpatient clinics in the Gampaha District of Sri Lanka from 01 November 2021 to 01 September 2022. The study methodology was previously reported in the BMJ Open [26]. Consecutive, consenting adults with type 2 diabetes(T2DM) and ultrasonographic (US) evidence of SLD detected within the previous 3 months were recruited to the study. We excluded patients with no consent, incomplete data to calculate FIB-4 scores, established cirrhosis on US scan, significant alcohol consumption (males > 14 units/week and females > 7 units/week), diagnosed liver diseases of known aetiology other than MASLD (e.g., autoimmune hepatitis, viral hepatitis, hemochromatosis, cholestatic liver disease, Wilson disease, etc.) and history of medication use known to cause SLD or liver fibrosis (e.g., tamoxifen, methotrexate etc.).

Trained medical graduates interviewed eligible patients, reviewed medical records, and collected data on demographics, metabolic risk factors, micro/macro vascular complications of diabetes (i.e., stroke, ischaemic heart disease, peripheral vascular disease, neuropathy, nephropathy or retinopathy), investigations, medications, diet, and exercise using an interviewer-administered questionnaire. We measured height, weight and waist circumference at recruitment and calculated body mass index (BMI) using the formula: weight (kg)/height (m)$^2$. Diabetes was defined according to the 2021 American Diabetes Association criteria, or if they were on medications, including insulin, for diabetes [27].

We calculated the FIB-4 in all as step 1, and those with a FIB-4 score ≥ 1.3 underwent VTCE as step 2 according to the European Association for the Study of the Liver (EASL) and American Diabetes Association guidelines [11,13]. We calculated the FIB-4 score using age and the most recent AST, ALT, and platelet count, which were obtained within 3 months of study recruitment, using the proposed formula [28]. A single, trained medical officer performed VCTE of all the patients in the study at the Gastroenterology and Hepatology unit of the Colombo North Teaching Hospital, Ragama, Sri Lanka. VCTE was performed using FibroScan® by Echosens machine, and data on liver stiffness measure (LSM) and controlled attenuation parameters (CAP) were recorded [29]. We defined the stage of liver fibrosis using LSM. SF was determined with an LSM of ≥8–12 kPa and ALD with an LSM of ≥12kPa [17]. All with a FIB-4 score <1.3 or LSM < 8 kPa were defined as having no SF.

We analysed data using IBM SPSS 22.0 software. We compared differences between patients with SF/ACLD and those without SF. Means of normally distributed continuous variables were compared using Student's t-test. Medians of non-normally distributed variables were compared using the Mann–Whitney U-test. Using multivariable logistic regression, we identified clinical factors that predict SF/ACLD by comparing patients with SF/ACLD and those without SF. Using the identified factors, we defined a new criterion for VCTE referral, in addition to using a FIB-4 score. We compared the number of patients needed to be referred for VCTE using FIB-4 only and the new criterion and calculated sensitivity, specificity, positive predictive value (PPV) and negative predictive value (NPV) of the two screening methods, i.e., FIB-4 score only as recommended by the EASL algorithm versus the new criterion in predicting SF/ACLD [30]. We compared the number of VCTE referrals that could be avoided by using the new criterion instead of using the FIB-4 score only as used in the EASL 2024 algorithm.

## External validation of a new criterion-based algorithm

We externally validated the new criterion-based algorithm in an external cohort of patients with T2DM and MASLD extracted from the VCTE database of the Colombo North Teaching Hospital (CNTH). We extracted all patients with diabetes diagnosed with MASLD who had complete data to calculate FIB-4 from those who underwent VCTE at CNTH from January 1, 2023, to September 22, 2025. We compared the number of individuals requiring referral for VCTE using the EASL-2024 algorithm and the new criterion-based algorithm. We studied the predictive accuracies of the two algorithms using confusion matrices. We determined the discriminative power of the algorithms using the area under the receiver operating characteristic curve (AUC-ROC).

Informed written consent was obtained from all participants before their recruitment for the study, and the study was conducted in accordance with the principles outlined in the Declaration of Helsinki. Ethical approval for the study was obtained from the Ethics Committee of the Faculty of Medicine, University of Kelaniya (P/66/07/2021).VCTE of the liver was done free of charge.

## Results

A total of 363 patients with T2DM and SLD were studied. The baseline characteristics of the study population are given in Table 1.

The patient flow in the study is shown in Fig 1.

Overall, 128 (35.3%) had a FIB-4 score ≥ 1.3, and 121 underwent VCTE. Seven were lost to follow-up. Of those who underwent VCTE, 76 (62.8%) had significant liver fibrosis or beyond, as diagnosed with an LSM ≥ 8.0 kPa, and 31 (4.4%) had advanced chronic liver disease (ACLD) defined by an LSM ≥ 12 (Table 2).

Associations of VCTE-diagnosed SF/ACLD are shown in Table 3. Patients with diabetes of ≥ 5 years' duration (OR 3.8, p = 0.035), micro/macrovascular complications (OR 19.4, p = 0.016), and a BMI of ≥ 25 kg/m² (OR 6.2, p = 0.003) had statistically significant associations with SF/ACLD on multivariable logistic regression. Association with age, ALT, AST, and platelet counts with SF/ACLD was not assessed separately, as those parameters are already included in the FIB-4 score, which was used to determine VTCE referrals.

The performance of the FIB-4 score alone (used in the EASL algorithm) and the new criterion in predicting SF/ACLD is shown in Table 4. Using the FIB-4 score alone, as recommended in the EASL algorithm 2024, 96 out of 363 patients were predicted to have SF/ACLD; therefore, they were recommended to undergo VCTE, and 67 of these patients actually had SF/ACLD. Using the new criterion, 83 patients were recommended to have VCTE, and 62 had SF/ACLD. The new criterion reduced 13/96 (13.5%) VCTE referrals compared to using the 2024 EASL algorithm with a sensitivity of 81.6%(62/76), specificity of 53.3%(24/45), PPV of 74.7%(62/83), NPV of 63.2%(24/38), positive likelihood ratio (LR+) of 1.75 and negative likelihood ratio (LR-) of 0.35. The 2024 EASL algorithm using FIB-4 only had a sensitivity of 88.2%(67/76), specificity of 35.6%(16/45), PPV of 69.8%(67/96), NPV of 64.0%(16/25), LR+ of 1.38 and LR- of 0.33.

The new criterion-based algorithm was validated in an external cohort of 372 patients (male – 156(41.9%), mean age 59 (SD 8.6) years) with diabetes and MASLD. Of them, 269 had SF on LSM using FibroScan. The number of patients requiring referral for VCTE using the new criterion-based algorithm was 197, and the same number was 233 with the EASL 2024 algorithm. The confusion matrices of the screening algorithms using the new criterion and the EASL criterion are shown in Fig 2. The predictive accuracies and discriminative power of the new criterion-based algorithm were as follows: sensitivity, 61.3%; specificity, 68.9%; positive predictive value (PPV), 83.8%; negative predictive value (NPV), 40.6%; and AUC-ROC, 0.71 (CI, 0.66–0.76). The same results using the EASL 2024 algorithm were sensitivity, 72.1%, specificity, 62.1%, PPV, 83.3%, NPV, 46.0% and AUC-ROC, 0.76(CI 0.71–0.81). The new criterion-based algorithm reduced the number of VCTE referrals needed by 36/233(15.5%) with a higher specificity of 68.9% compared to 62.1% with the EASL 2024 algorithm.

**Table 1. Baseline characteristics of the study population.**

| | Total |
|---|---|
| | **n = 363** |
| Mean age, (SD), years | 53.6 (11.4) |
| Age ≤ 65 years, n(%) | 298 (82.1) |
| Male, n(%) | 187 (51.5) |
| **Ethnicity, n (%)** | |
| Sinhalese | 343 (94.5) |
| Other ethnicities | 20 (5.5) |
| Current smokers, n (%) | 24 (6.6) |
| Type 2 diabetes mellitus, n (%) | 363 (100.0) |
| Duration of diabetes, median (IQR) years | 5 (2 – 10 ) |
| Diabetic complications present*, n (%) | 65 (17.9) |
| Hypertension, n (%) | 192 (52.9) |
| Dyslipidaemia, n (%) | 247 (68.0) |
| Ischaemic heart disease, n (%) | 26 (7.2) |
| Family history of liver disease, n (%) | 51 (14.0) |
| History of hepatitis B, n (%) | 0 (0.0) |
| History of hepatitis C, n (%) | 0 (0.0) |
| Alcohol intake, n(%) | |
| ethanol > 30 g/day in men | 0 (0.0) |
| ethanol > 20 g/day in women | 0 (0.0) |
| Body mass index (kg/m$^2$) | |
| < 18.5 (Underweight) | 3(0.8) |
| 18.5–24.9 (Normal Weight) | 128(35.3) |
| 25.0–29.9 (Overweight) | 154(42.4) |
| ≥ 30.0 (Obesity) | 78(21.5) |
| Waist circumference (cm) | |
| Mean, SD in females | 89.9(8.99) |
| Mean, SD in males | 90.4(7.85) |
| Abdominal obesity present**, n (%) | 266(73.3) |
| **Ultrasound scan grading of steatotic liver disease** | |
| I | 152(41.9) |
| II | 164(45.2) |
| III | 47(12.9) |
| IV | 0(0.0) |

\* stroke, ischaemic heart disease, peripheral vascular disease, neuropathy, nephropathy or retinopathy, \*\* waist ≥80 cm in females, ≥ 90 cm in males,

n-number, SD-standard deviation, IQR-interquartile range, FIB-4 fibrosis-4, VTCE-vibration-controlled transient elastography

## Discussion

This study identified independent risk associations for SF/ACLD in a cohort of Sri Lankans with diabetes and MASLD. We then developed a new criterion to prioritise VCTE referrals for investigating SF/CLCD in MASLD and compared it with using only the FIB-4 score, as recommended in the EASL 2024 algorithm. The new criterion uses the FIB-4 cutoffs used in the EASL 2024 algorithm (FIB-4 ≥ 1.3 in ≤65 year-olds and ≥2.0 in > 65 year-olds) with one or more of three other factors,

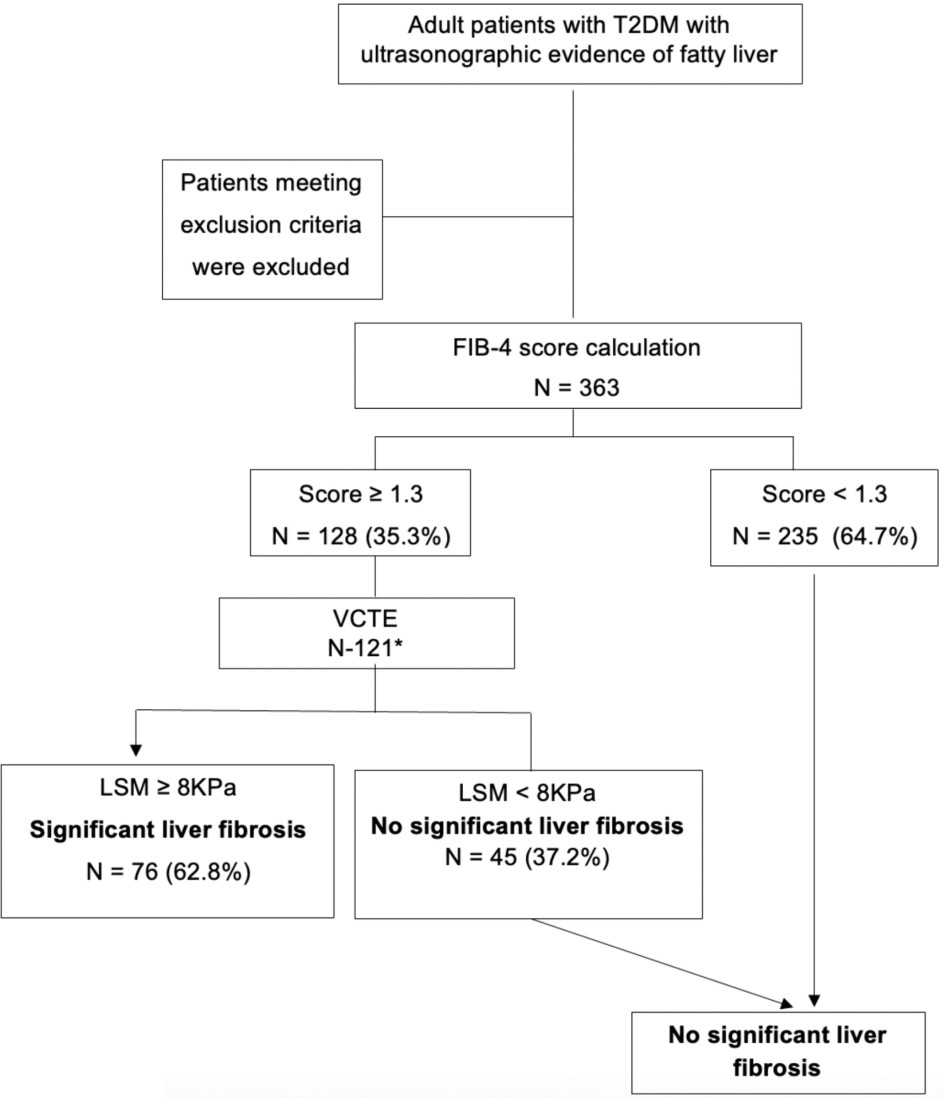

**Fig 1. Patient flow through the study.** T2DM type 2 diabetes mellitus, USS ultrasound scan, FIB-4 Fibrosis-4, N number, VTCE vibration-controlled transient elastography, and LSM liver stiffness measure. * lost to follow-up n = 7.

i.e., diabetes of ≥ 5 years duration, presence of micro/macrovascular complications or BMI ≥ 25 kg/m². We showed that adopting the new criterion could reduce VCTE referrals by 13.5% and the findings were externally validated. In ruling in SF, a patient selected to have VCTE using the new criterion was almost twice as likely to have SF, whereas it was only 1.38 times more likely when using the FIB-4 score. However, ruling out SF was similar with both the new criterion and the FIB-4 score, where two out of three persons not selected for VCTE were unlikely to have SF.

The associations we found are in keeping with the literature from the West. SF/ACLD is significantly associated with older age, obesity, diabetes mellitus, high ALT, hypertriglyceridemia, insulin resistance, a high waist-to-hip ratio, and a low platelet count, some of which are incorporated into the FIB-4 score [31].

There are epidemiological differences in MASLD among South Asians compared to White Caucasians [32,33]. A study from Hong Kong and Malaysia observed that only 20% of patients referred for VCTE using the current guidelines

**Table 2. Measures of liver fibrosis in the study sample.**

|  | N (%) |
|---|---|
| **FIB-4 score (N = 363)** |  |
| < 1.3, n(%) | 235(64.7) |
| 1.3–2.67, n(%) | 112(30.9) |
| > 2.67, n(%) | 16(4.4) |
| **VCTE indicated according to EASL 2024** | 96 |
| Age ≤ 65y and FIB-4 ≥ 1.3 | 80 |
| Age > 65y and FIB-4 ≥ 2.0 | 16 |
| **Liver stiffness measure using VTCE -, *kPa* (N = 121\*)** |  |
| Median, (IQR) | 7.4(5.8-10.0) |
| < 8, n(%) (no significant fibrosis) | 45(37.1) |
| ≥ 8–11.9, n(%) (significant fibrosis) | 45(37.1) |
| ≥ 12, n(%) (advanced fibrosis) | 31(25.6) |

missing data; * 7

n-number, IQR-Interquartile range, FIB-4 Fibrosis-4, VTCE- vibration-controlled transient elastography

EASL- European Association for the Study of the Liver

**Table 3. Associations of significant fibrosis of the liver.**

|  | SF/ ACLD (N = 29) | No SF (N = 67) | Univariate analysis P | Multivariable analysis | | |
|---|---|---|---|---|---|---|
|  |  |  |  | *OR* | *95% CI* | *P\** |
| Male sex | 15 | 39 | 0.556 | 0.32 | 0.09-1.14 | 0.080 |
| Hypothyroidism | 2 | 10 | 0.275 | 3.00 | 0.3-29.55 | 0.347 |
| Ischaemic heart disease | 1 | 6 | 0.341 | 1.89 | 0.12-29.52 | 0.649 |
| Hypertension | 16 | 32 | 0.505 | 0.43 | 0.13-1.38 | 0.155 |
| Dyslipidaemia | 22 | 46 | 0.476 | 0.60 | 0.17-2.04 | 0.409 |
| Diabetes for ≥5 years | 11 | 39 | 0.068 | 3.80 | 1.1-2.04 | **0.035** |
| Micro/macro-vascular complications | 1 | 16 | **0.016** | 19.40 | 1.79-210.66 | **0.015** |
| Current smoking | 3 | 5 | 0.639 | 0.27 | 0.03-2,79 | 0.269 |
| Family history of liver disease | 4 | 14 | 0.413 | 3.94 | 0.71-21.84 | 0.117 |
| SLD≥ grade II on USS | 6 | 22 | 0.229 | 4.09 | 0.94-17.86 | 0.061 |
| Risk diet | 12 | 30 | 0.758 | 0.70 | 0.21-2.28 | 0.553 |
| Sedentary lifestyle | 21 | 42 | 0.357 | 0.51 | 0.14-1.89 | 0.316 |
| BMI ≥ 25 kg/m$^2$ | 14 | 51 | **0.007** | 6.21 | 1.83-21.02 | **0.003** |

* Multivariable logistic regression adjusted for all the variables mentioned in the table

n-number, IQR-Interquartile range, CI-Confidence interval

SF-Significant Fibrosis, ACLD-Advanced Chronic Liver Disease, FIB-4 Fibrosis-4, VTCE- vibration-controlled transient elastography, BMI- Body Mass Index, USS-Ultrasound scan

SF/ACLD significant fibrosis/advanced chronic liver disease, VTCE – vibration-controlled transient elastography, OR odds ratio, CI – Confidence Interval, p- probability, USS ultrasound scan, BMI- body mass index

Based on the above data (Table 3), we defined a new criterion for VCTE referrals; I.e., "having a FIB-4 score ≥1.3 in ≤65-year-olds or FIB-4 score ≥2 in >65-year-olds and having one or more of three other criteria: diabetes ≥5 years duration, presence of diabetic micro/macrovascular complications, or BMI ≥ 25 kg/m$^2$.

**Table 4. Performance of Fib-4 score alone and the new criterion in predicting significant fibrosis.**

| | VCTE diagnosis | | | |
| --- | --- | --- | --- | --- |
| | | Significant fibrosis | No significant fibrosis | Total |
| **Using FIB-4 score** | VCTE indicated | 67 | 29 | 96 |
| | VCTE not indicated | 9 | 16 | 25 |
| | Total | 76 | 45 | 121 |
| **Using the new criterion** | VCTE indicated | 62 | 21 | 83 |
| | VCTE not indicated | 14 | 24 | 38 |
| | Total | 76 | 45 | 121 |

FIB-4 fibrosis-4, VTCE- vibration-controlled transient elastography

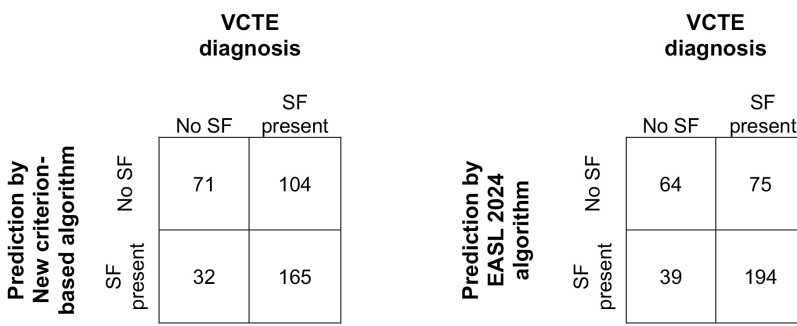

**Fig 2. Predictive accuracies of algorithms using confusion matrices.** EASL European Association for the Study of the Liver, SF significant fibrosis.

had AF [4]. The genetic and epidemiological differences between Asians and Caucasian whites explain the factors used in our new criterion for predicting SF/ACLD. Asians develop complications of diabetes earlier than the population in the West, with a genetic predisposition [34,35]. The BMI cutoff for overweight in Asians (23 Kg/m²) is lower than for white Caucasians [36,37]. Further, Asians do have more abdominal obesity compared to the White Caucasians and therefore, South Asians with relatively lower BMIs are also at high CV risk.

The new criterion we developed is simple, freely available and suited for low-resource settings. A cost-effective screening tool should be able to detect more patients with SF/ACLD while doing a limited number of VCTEs. Our new criterion, which reduced referrals by 13.5%, had a 75% PPV and 53% specificity, compared to 70% PPV and 36% specificity when using only the FIB-4 score.

Our study has several strengths. All data were prospectively collected. All VCTEs were performed using a single machine by a single operator to minimise interrater variability. We used data only from patients who underwent VCTE to identify factors associated with SF/ACLD. However, there are limitations. Although liver biopsy is the gold standard for staging liver fibrosis, we used VCTE to define SF/ACLD [38]. However, this is the current practice worldwide, and liver biopsy for SLD is rarely performed because of its invasive nature [31,39]. According to the EASL 2024 algorithm, we did VCTE only in patients with high FIB-4 scores above the screening cutoffs and identified associations by studying their data. However, as around 10% of patients with SF/ACLD could be missed using the FIB-4 cutoff in the first screening step, this could have led to a selection bias. As we studied only a cohort of Sri Lankans with diabetes and MASLD, our results are not generalisable to all low-resource settings.

In conclusion, the new referral criterion will help reduce the number of VCTE referrals to detect SF/ACLD in resource-limited settings.

## Author contributions

**Conceptualization:** Chamila Mettananda.

**Data curation:** Chamila Mettananda, Thimira Egodage, Channaka Dantanarayana, Rumal Fernando, Lakmali Ranaweera, Arunasalam Pathmeswaran.

**Formal analysis:** Chamila Mettananda, Thimira Egodage, Channaka Dantanarayana, Arunasalam Pathmeswaran.

**Funding acquisition:** Chamila Mettananda.

**Investigation:** Chamila Mettananda, Chamila Ranawaka, Thimira Egodage, Channaka Dantanarayana, Rumal Fernando, Lakmali Ranaweera, Dulani Kottahachchi, Shirom Siriwardhana, Anuradha Dassanayake, Janaka de Silva.

**Methodology:** Chamila Mettananda, Chamila Ranawaka, Rumal Fernando, Arunasalam Pathmeswaran, Anuradha Dassanayake, Janaka de Silva.

**Project administration:** Chamila Mettananda, Lakmali Ranaweera.

**Resources:** Chamila Mettananda, Chamila Ranawaka, Lakmali Ranaweera, Dulani Kottahachchi, Shirom Siriwardhana, Anuradha Dassanayake, Janaka de Silva.

**Software:** Chamila Mettananda.

**Supervision:** Chamila Mettananda, Chamila Ranawaka.

**Validation:** Chamila Mettananda.

**Visualization:** Chamila Mettananda.

**Writing – original draft:** Chamila Mettananda.

**Writing – review & editing:** Chamila Mettananda, Chamila Ranawaka, Thimira Egodage, Channaka Dantanarayana, Rumal Fernando, Lakmali Ranaweera, Dulani Kottahachchi, Shirom Siriwardhana, Arunasalam Pathmeswaran, Anuradha Dassanayake, Janaka de Silva.

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
