## [Decision Letter · Decision Letter 0]

29 Dec 2025

Dear Dr. Mettananda,

Thank you for submitting your manuscript to PLOS ONE. After careful consideration, we feel that it has merit but does not fully meet PLOS ONE’s publication criteria as it currently stands. Therefore, we invite you to submit a revised version of the manuscript that addresses the points raised during the review process.

Dear Authors, the manuscript entitled "Adoption and validation of the European Association for the Study of the Liver algorithm for the noninvasive diagnosis of advanced fibrosis in metabolic dysfunction-associated steatotic liver disease in low-resource South Asian settings", is a reasonable proposal to use resources in clinical scenarios where resources are limited, however it needs some adjusments:

- In table 1, complete all parenthesis

- In table I, use the universal recognized categories for BMI according to cut-off values. You may need to recalculate the frequencies.

- In table I, divide the waist by sex and treat it as a numeric discrete variable. Waist itself is related to liver steatosis and could be a great prognosis factor to MASLD, however it requieres a better statistical treatment, I suggets that you could consider it.

- Explain the importance to divide by ethnic groups, if there is no another analysis with this variable, is enough to stablish "Sinhalese" and "other ethnicities" (Tamil, Muslim, Burgher, etc.)

- Describe the reasons why 7 participants were lost.

We look forward to receiving your revised manuscript.

Kind regards,

Sophia Eugenia Martínez-Vázquez, Ph.D.

Academic Editor

PLOS One

Journal Requirements:

“This study was supported by the Grant RC/2024/03  of the University of Kelaniya, Sri Lanka, awarded to Chamila Mettananda.”

5. We note that you have indicated that there are restrictions to data sharing for this study. For studies involving human research participant data or other sensitive data, we encourage authors to share de-identified or anonymized data. However, when data cannot be publicly shared for ethical reasons, we allow authors to make their data sets available upon request. For information on unacceptable data access restrictions, please see http://journals.plos.org/plosone/s/data-availability#loc-unacceptable-data-access-restrictions.

6. Please ensure that you refer to Figure 2 in your text as, if accepted, production will need this reference to link the reader to the figure.

Reviewers' comments:

Reviewer's Responses to Questions

**Comments to the Author**

1. Is the manuscript technically sound, and do the data support the conclusions?

Reviewer #1: Yes

Reviewer #2: Yes

2. Has the statistical analysis been performed appropriately and rigorously?

Reviewer #1: Yes

Reviewer #2: Yes

3. Have the authors made all data underlying the findings in their manuscript fully available?

Reviewer #1: Yes

Reviewer #2: Yes

4. Is the manuscript presented in an intelligible fashion and written in standard English?

Reviewer #1: Yes

Reviewer #2: Yes

Reviewer #1: Dear authors, Thank you for this work. Here are my comments for more clarity:

# Refine the introduction and problem statement: While the introduction effectively highlights the burden of MASLD and T2DM, consider strengthening the rationale for developing a new criterion for VCTE referral. Emphasize why the existing EASL algorithm might be insufficient or impractical in low-resource South Asian settings, beyond just reducing referrals.

# Patient selection: Provide more details on the inclusion and exclusion criteria for both the primary and external validation cohorts. For instance, were patients with other liver diseases excluded?

# AUC-ROC values: While AUC-ROC values are provided, consider including confidence intervals for these values to indicate the precision of the estimates.

Reviewer #2: 1.The manuscript is technically sound. The study design is appropriate for validating a modified algorithm, and the data generally support the conclusions: multivariable logistic regression identifies independent predictors of SF/ACLD, and the new criterion reduces VCTE referrals while maintaining reasonable sensitivity (81.6% vs. 88.2% for EASL) and improving specificity (53.3% vs. 35.6%). The external validation strengthens the findings.

2.The statistical methods are appropriate for the study design. The use of multivariable logistic regression to identify independent predictors is standard. Performance metrics (sensitivity, specificity, PPV, NPV) and AUC-ROC for the algorithms are correctly calculated and presented. The comparison of referral numbers between the original and modified algorithms is straightforward and valid. The analysis is clearly described in the methods.

3.The authors state that data cannot be shared publicly due to ethical restrictions but are available from the corresponding author for researchers meeting criteria for access to confidential data. This is an acceptable data availability statement for clinical datasets containing potentially identifiable patient information.

4.The manuscript is well-presented, logically structured (abstract, introduction, methods, results, discussion), and written in clear, standard English.

**Do you want your identity to be public for this peer review?** For information about this choice, including consent withdrawal, please see our Privacy Policy

Reviewer #1: No

Reviewer #2: **Yes:** Zheng Xu

---

## [Author Response · Author response to Decision Letter 1]

4 Jan 2026

Response to reviewer comments:

1. Comment: In table 1, complete all parentheses

Response: Thank you for picking it. We corrected all.

2. Comment: In table I, use the universal recognized categories for BMI according to cut-off values. You may need to recalculate the frequencies. Thank you.

Response: We recalculated and included it.

3. Comment: In table I, divide the waist by sex and treat it as a numeric discrete variable. Waist itself is related to liver steatosis and could be a great prognosis factor to MASLD, however it requieres a better statistical treatment, I suggets that you could consider it.

Response: Thank you for the constructive comment. In the baseline characteristics, we demonstrated abdominal obesity, as classified universally, in both men and women. Now, in the revised version, we have shown the mean waist in men and women separately, as you rightly pointed out. It was normally distributed.

4. Comment:Explain the importance to divide by ethnic groups, if there is no another analysis with this variable, is enough to stablish "Sinhalese" and "other ethnicities" (Tamil, Muslim, Burgher, etc.).

Response: Thank you. Agree with your comment. We do see a lot of Muslim people having steatotic liver disease, but when we separately analysed ethnicity as an associated factor for SF, it did not show any significant association. Probably it is due to Muslim people having high BMIs. So changed the ethnicity as you suggested.

5. Comment: Describe the reasons why 7 participants were lost.

Response: These patients did not attend the fibroscanning appointment despite receiving reminders.

6. Comment: In the discussion section, please state the real utility of your new criterion for VCTE using different frequencies of the disease, and please give examples of the likelihood ratio.

Response: Thank you for the comment to improve the paper. We calculated positive/negative likelihood ratios (lines 234-237) and strengthened our discussion using these results (lines 269-273).

---

## [Editor Report · Decision Letter 1]

6 Jan 2026

Adoption and validation of the European Association for the Study of the Liver algorithm for the noninvasive diagnosis of advanced fibrosis in metabolic dysfunction-associated steatotic liver disease in low-resource South Asian settings.

PONE-D-25-61395R1

Dear Dr. Mettananda,

We’re pleased to inform you that your manuscript has been judged scientifically suitable for publication and will be formally accepted for publication once it meets all outstanding technical requirements.

Kind regards,

Sophia Eugenia Martínez-Vázquez, Ph.D.

Academic Editor

PLOS One

Additional Editor Comments (optional):

Dear Authors,

Thank you for the reception and settings you made. I consider the manuscript is ready. Congratulations.
---

## [Editor Report · Acceptance letter]

PONE-D-25-61395R1

PLOS One

Dear Dr. Mettananda,

I'm pleased to inform you that your manuscript has been deemed suitable for publication in PLOS One. Congratulations! Your manuscript is now being handed over to our production team.

Kind regards,

on behalf of

Dr. Sophia Eugenia Martínez-Vázquez

Academic Editor

PLOS One